# Effects of Pneumococcal Vaccination in Children Under Five Years of Age in the Democratic Republic of Congo: A Systematic Review

**DOI:** 10.3390/vaccines13060603

**Published:** 2025-05-31

**Authors:** Marcellin Mengouo Nimpa, Abel Ntambue, Christian Ngandu, M. Carolina Danovaro-Holliday, André Bita Fouda, Aimé Mwana-Wabene Cikomola, Jean-Crispin Mukendi, Dieudonné Mwamba, Adèle Daleke Lisi Aluma, Moise Désiré Yapi, Jean Baptiste Nikiema, Boureima Hama Sambo, Daniel Katuashi Ishoso

**Affiliations:** 1World Health Organization (WHO) Country Office, Kinshasa P.O. Box 1899, Democratic Republic of the Congo; nimpamengouom@who.int (M.M.N.); yapimo@who.int (M.D.Y.); nikiemaje@who.int (J.B.N.); sambob@who.int (B.H.S.); 2École de Santé Publique, Université de Lubumbashi, Lubumbashi P.O. Box 1825, Democratic Republic of the Congo; abelntambue@gmail.com; 3National Institute of Public Health, Kinshasa P.O. Box 01206, Democratic Republic of the Congo; nganduchristian@ymail.com (C.N.); dk.mwamba@umontreal.ca (D.M.); 4Immunization, Analytics and Insights (IAI), Department of Immunization, Vaccines and Biologicals (IVB), World Health Organization (WHO), 1211 Geneva, Switzerland; 5World Health Organization African Regional Office, Brazzaville P.O. Box 06, Congo; abita@who.int; 6Expanded Program of Immunization, Kinshasa P.O. Box 01206, Democratic Republic of the Congo; aimcik@yahoo.fr (A.M.-W.C.); mukendijean2@gmail.com (J.-C.M.); 7Independent Researcher, Kinshasa P.O. Box 01211, Democratic Republic of the Congo; alumaadele@yahoo.fr; 8Kinshasa School of Public Health, University of Kinshasa, Kinshasa P.O. Box 11850, Democratic Republic of the Congo

**Keywords:** pneumococcal conjugate vaccine, PCV13, pneumococcal infections, *Streptococcus pneumoniae*, Democratic Republic of the Congo

## Abstract

Background: In the Democratic Republic of Congo (DRC), the 13-valent pneumococcal conjugate vaccine (PCV13) was introduced in 2011 through a three-dose schedule, targeting infants as part of the Expanded Program on Immunization (EPI), to reduce pneumococcal-related morbidity and mortality. The aim of this study was to determine the proportion of pneumonia and meningitis cases and deaths prevented in children under five following the introduction of this vaccine. Methods: This is a systematic review. We synthesized findings from studies carried out in the DRC between 2011 and 2023. We searched scientific articles, published and unpublished doctoral theses and conference proceedings. Only papers written in French or English and those reporting the results of original analytical studies were selected. We assessed the direct effect of PCV13 by calculating the proportion of infections avoided, using Odds Ratios or prevalence ratios related to infection or pneumococcal carriage. Results: Four studies were included in this review. Regarding pneumococcal carriage, when children received three PCV13 doses, the prevalence of carriage was reduced by 93.3% (95% CI: 86.3 to 96.6%), while a single dose did not significantly reduce the prevalence of carriage compared with children who had not received any dose. Concerning pneumonia prevention, three doses of PCV13 prevented 66.7% (95% CI: 37.2 to 82.2) of cases among vaccinated children. The proportion of meningitis attributable to *S. pneumoniae* prevented was 75.0% (95% CI: 6% to 93.3%) among children vaccinated with PCV13. *S. pneumoniae* serotypes 19F and 23F were the most frequent causes of invasive pneumonia in children. Serotypes 35B/35C, 15B/C, 10A and 11A/D were the most frequently identified causes of morbidity in Congolese children. In 2022, with PCV13 vaccination coverage at 79.0%, an estimated 113,359 cases of severe pneumonia and 17,255 pneumonia-related deaths were prevented in the DRC, with 3313 cases and 1544 deaths attributable to pneumococcal meningitis prevented. Conclusions: There is clear, but scattered, evidence of reduced colonization by *S. pneumoniae* and hospital admissions due to pneumococcal pneumonia and meningitis. The results also show that *S. pneumoniae* serotypes 35B/35C, 15B/C, 10A and 11A/D not included in PCV13 were the main cause of pneumococcal disease in unvaccinated or under-vaccinated children. These data support the need to continue improving vaccination coverage among children who are unvaccinated or incompletely vaccinated with PCV13 to reduce the burden of pneumococcal infections in the DRC.

## 1. Introduction

*Streptococcus pneumoniae*, the bacterium responsible for pneumococcal infections, can lead to severe invasive diseases such as pneumonia, meningitis, septicemia, and acute otitis media, especially in children under two years of age. These infections are particularly prevalent among young children and older adults [1,2].

In most cases, pneumococcal illness is preceded by silent colonization of the nasopharynx. Globally, *S. pneumoniae* is estimated to cause over 318,000 deaths annually (with an uncertainty range of 207,000 to 395,000) among children aged 1 to 59 months, with Africa bearing the greatest share of this mortality burden [3].

Prior to the inclusion of pneumococcal conjugate vaccines (PCVs) in national immunization schedules, more than 70% of invasive pneumococcal diseases were linked to serotypes targeted by these vaccines [3]. Pneumonia represents the most common clinical form of infection caused by this pathogen [4]. Meningitis, although less frequent, accounts for about 2% of serious pneumococcal cases and contributes to 12% of related fatalities [3,5].

Before vaccine introduction, nasopharyngeal carriage rates in children under five in low- and middle-income countries ranged from 20% to 90%, making them the principal reservoir and source of pneumococcal transmission [6].

Over 90 distinct pneumococcal serotypes have been identified, each exhibiting varying degrees of virulence and antimicrobial resistance [5,6,7]. The management of these infections has become increasingly challenging due to the global emergence of strains resistant to penicillin and other antibiotics [8].

Pneumococcal vaccines target the serotypes most often involved in invasive infections. There are two types; the first is a conjugate vaccine (PCV13) and contains 13 serotypes 1, 3, 4, 5, 6A, 6B, 7F, 9V, 14, 19A, 19F, 18C and 23F. It provides protection against pneumococcal infection and carriage [1]. Moreover, it can be used from the age of six weeks. By reducing carriage, it provides an indirect (herd) protection effect: unvaccinated children are also less likely to be infected by pneumococcus [9]. The second vaccine is unconjugated and contains 23 pneumococcal serotypes (1, 2, 3, 4, 5, 6B, 7F, 8, 9N, 9V, 10A, 11A, 12F, 14, 15B, 17F, 18C, 19A, 19F, 20, 22F, 23F and 33F). Although this vaccine has a broader spectrum, it is not effective before the age of two, does not eliminate nasopharyngeal carriage—the primary source of person-to-person transmission—provides only short-lived protection, and lacks a booster response upon revaccination [9].

In countries where PCV vaccination is widespread, the number of cases of meningitis and pneumococcal pneumonia in children under the age of five has fallen markedly [10,11]. An impact has also been observed in unvaccinated children, adults, and the elderly owing to herd immunity conferred by childhood vaccination. In addition, vaccination has led to a decline in the rate of antibiotic-resistant pneumococci in meningitis, bacteremia, and pneumococcal otitis [12].

In the Democratic Republic of Congo (DRC), studies show that pneumonia is the second leading cause of death in children under the age of five, after malaria [13]. This bacterium is also responsible for 28.5% of purulent meningitis [14]. In 2011, pneumococcus was responsible for 41,939 deaths attributable to pneumonia and 9256 deaths attributable to meningitis in children under five [15].

From April 2011, PCV13 was gradually introduced into the DRC’s Expanded Program on Immunization (EPI). The vaccination schedule includes three doses of the vaccine: the first at 6 weeks after birth, the second at 10 weeks, and the third at 14 weeks. However, children can also be vaccinated at any contact for catch-up if the routine schedule has not been followed.

In 2011, when PCV13 was introduced into the EPI, under-five mortality was 104 per 1000 live births [16]. It was therefore accepted that vaccination should help to substantially reduce the burden of morbidity and mortality in this age group [16]. The estimated PCV13 coverage has increased from 9.0% in 2011 to 79.0% in 2022 [15], but we do not know, in the specific context of the DRC, what effect this vaccine has had on the burden of disease preventable by PCV13, particularly regarding the number of severe pneumonia and meningitis cases and related deaths averted.

In the DRC, the scientific literature on the effects of PCV13 is limited [17,18,19,20,21,22,23,24,25,26,27,28]. What evidence is available comes from a few studies carried out in limited geographical areas: children presenting different clinical pictures of pneumococcal carriage [26], invasive pneumonia [19], and pneumococcal meningitis [22,27] with different inclusion criteria from one study to another. These results, fragmented and localized, have not yet been pooled to determine the effects of the introduction of PCV13 on the burden of pneumococcal infections in the DRC.

The aim of this study was to determine, by synthesizing the available scientific literature on PCV13 vaccination in the DRC, the fraction of cases of pneumonia and meningitis and the fraction of deaths in children under five attributable to these diseases prevented by this vaccine in the DRC, and to determine the evolution of isolated *S. pneumoniae* serotypes since the introduction of PCV13.

## 2. Materials and Methods

This systematic review was carried out in line with the PRISMA group’s recommendations. A research librarian contributed to the development of the search strategy and the inclusion criteria. The review protocol was registered in the International Prospective Register of Systematic Reviews (PROSPERO) under the registration number CRD42025641327 (https://www.crd.york.ac.uk/prospero/#recordDetails, accessed on 21 January 2020).

### 2.1. DRC Context

The DRC is one of the most densely populated countries in Africa. In 2022, its population was estimated at 105 million, occupying an area of 2,345,000 km^2^, with a density of 45 inhabitants per km^2^, according to the Statistical Yearbook of the Ministry of Planning of the DRC. Children under the age of five represented 20% of the total population.

In 2009, two years prior to the rollout of PCV13, pneumococcal infections affected more than 3000 out of every 100,000 children under five years old, with a mortality rate ranging between 300 and 500 per 100,000 in the same age group [29]. By 2022, under-five mortality was recorded at 98 deaths per 1000 live births, and pneumococcal disease was responsible for 16.0% of these fatalities, based on data from Countdown2030 [13].

Based on vaccination coverage surveys, and using a linear interpolation model with Lives Saved Tool (LiST) software, version 6.0, PCV13 vaccination coverage between 2011 and 2022 varied from 9.0% to 79.0% (Figure 1) [30].

### 2.2. Study

This study is a systematic review of the literature on the effects of PCV13 (3 doses) and the evolution of *S. pneumoniae* serotypes since the introduction of this vaccine in the EPI in the DRC. We assessed the effect of the vaccine by summarizing the effects reported in the literature from scientific research conducted in the DRC and published from 2011 to 2023.

### 2.3. Documentary Search Strategy

We searched for scientific articles in online journals, local offline journals, published and unpublished Ph.D. theses, and proceedings of scientific congresses and symposia.

To locate pertinent studies, we conducted a search across the Medline, Embase, Cochrane Library, and Google Scholar databases, utilizing Medical Subject Headings (MeSH), controlled vocabulary, and free-text keywords. Additionally, we used keyword-based searches in PubMed to capture articles and publications not indexed in Medline. To ensure comprehensive coverage, we also performed a manual review of reference lists from relevant systematic reviews to identify any potentially overlooked studies.

In these bibliographic databases, the keywords used were as follows: “13-valent pneumococcal vaccine AND Democratic Republic of the Congo”; “13-valent pneumococcal vaccine AND Africa”; “PCV13 and *Streptococcus pneumoniae* serotypes”; “PCV13 and invasive pneumonia in the Democratic Republic of the Congo”; “PCV13 and meningitis in the Democratic Republic of the Congo”.

After conducting all the bibliographic searches on all published and unpublished documents on the DRC or in Africa, reporting data on the impact of PCV13 vaccination on the incidence of pneumonia and pneumococcal meningitis, we organized working meetings with experts in the field of bacterial disease surveillance to search for additional reports to ensure the completeness of the reports collected that could be taken into account in this systematic review.

We used the Effective Public Health Practice Project’s tool for assessing the quality of quantitative studies to evaluate the level of risk of bias in each study, as well as its internal validity [31]. Each study was assessed against eight methodological criteria: selection bias, study design, confounding factors, blinding if it was a clinical trial, data collection methods, study participant withdrawals, intervention integrity and analysis.

We then assessed the collected papers, with two reviewers first examining the titles and abstracts, and then the full text. Disagreements about inclusion were resolved by discussion between the reviewers.

### 2.4. Inclusion Criteria

We included only papers written in French or English and those reporting original research results. When meta-analyses included DRC data, we searched for the original article. Where this was not found, the meta-analysis was not included in this review.

This review considered a range of study designs, including randomized controlled trials, non-randomized studies, analyses based on surveillance data, and observational research. Titles and abstracts underwent two rounds of screening, and those addressing at least one of the four targeted outcomes—carriage, invasive disease, pneumonia or meningitis, and indirect effects—were subjected to a detailed review using a purpose-designed data extraction tool.

### 2.5. Measuring the Effects of PCV13

#### 2.5.1. Measures from Studies

The studies included in this review determined the direct effect of vaccination by calculating Odds Ratios (ORs) or prevalence ratios of infection or carriage in vaccinated children (PCV13) compared with those who were not vaccinated (non-PCV13) [18,23,28].

We did not carry out meta-analyses because of the heterogeneity between studies. From the ORs, we calculated the fraction of infections prevented (FIP) in PCV13 children according to the formula provided by Bouyer C et al. if the ORs were of less than 1 [32]. However, when the ORs were greater than 1, we used the formula provided by the Global Burden of Disease (GBD) 2016 Lower Respiratory Infections Collaborators [4].

#### 2.5.2. Estimation of the Burden of Pneumococcal Infections Prevented in the DRC

We modeled the expected direct effect of the vaccine as a function of the effect of the vaccine reported in the studies included in this review and its coverage in the DRC population. We derived the total effect of PCV13 on different levels of vaccine coverage from a report modeling the cohort effect in India [33].

To estimate the number and percentage of cases and deaths attributable to pneumonia and meningitis caused by *S. pneumoniae* prevented and averted by PCV13, we established the effect of this vaccine on the specific causes of morbidity and mortality in children under five years of age. The effects of the change in PCV13 vaccination coverage were applied to the number of cases and deaths in children under five years of age as documented since 2000 for DRC, on the assumption that this effect on the population increased linearly with vaccination coverage [30]. The following formula was used to calculate the number of infections prevented and deaths avoided in the population (FIPp).$$Deaths\;averted=\frac{N\times I\times (P1-P0)}{\left(1-I\times P0\right)}$$

N = number of cases or deaths with vaccination coverage in the initial year (2011) and the year before the year under evaluation.

I = percentage by which PCV13 prevents new cases or reduces deaths.

P0 = vaccination coverage in the initial year (2011) and the year before the year under evaluation.

P1 = vaccination coverage in the year evaluated.

We consider the herd immunity in the model as the additional proportion of cases or deaths due to vaccine-susceptible pneumococci that could be prevented by population-level PCV coverage, in addition to the cases and deaths prevented directly by vaccination.

Analyses relating to FIP were carried out using NCSS 2020 software, while calculations of cases prevented and deaths avoided were carried out using the LiST tool in OneHealth software v6.29 [30].

## 3. Results

### 3.1. Selection of Studies

In total, we identified eleven articles published in online scientific journals and one doctoral thesis available online relating to pneumococcal vaccination in the DRC (Figure 2).

We excluded four articles (33.3%) that were either systematic reviews or studies based on evaluations using mathematical models. Of the eight studies based on primary data, we excluded two descriptive studies that focused either on the prevalence of pneumococcal disease or on PCV13 vaccination coverage.

Finally, we excluded two articles due to methodological limitations, such as the absence of defined judgment criteria or undetermined vaccination status in children. Only four studies were included in this review.

### 3.2. Profile of Studies Included in This Review and Effects of PCV13

Table 1 shows that two of these three studies were carried out in the province of South Kivu in a hospital setting; the third study was conducted simultaneously in Kinshasa (Kalembelembe and Kingasani), and in Lubumbashi, at Sendwe Hospital. The first study [26] evaluating the effect of PCV13 on the burden of pneumococcal infections was carried out in 2013 and published in 2018, i.e., two years after the introduction of PCV13 in the national EPI schedule; the second was carried out in 2016 [19], i.e., five years after, and the third [27] was carried out in 2023, i.e., 12 years after the introduction of PCV13. Only the study by Coulibaly et al. used data from sentinel surveillance sites for meningitis and pneumococcal infections.

The first study [26] was cross-sectional. In this study, the authors compared *S. Pneumoniae* carriage in PCV13 recipients (2–3 doses and 1 dose) versus non-PCV13 children under five years of age. Coulibaly et al. [19] used a quasi-experimental study in which they compared the occurrence of pneumococcal infections between children in a province that had already introduced PCV13 (Kinshasa) and those in another province that had not yet done so (Katanga). The third study [27], conducted at Panzi Hospital in Bukavu, was also cross-sectional and was conducted among children admitted to pediatric wards with confirmed meningitis. It compared the presence of *S. pneumoniae* as a cause of meningitis between PCV13-positive children and those who were not.

In the studies included in this review, the burden of pneumococcal infections was assessed mainly by calculating ORs. As shown in Table 1, the proportion of children with *S. pneumoniae* pneumonia was six times lower in children vaccinated with PCV13 (3 doses) than in those who were not. In the study by Coulibaly et al., the incidence of invasive pneumonia was five times higher in non-PCV13 than in PCV13 children (95% CI: 0.03–0.14). The study by Manegabe et al. showed that non-PCV13 children had four times the odds of suffering from pneumococcal meningitis than PCV13 children (three doses).

Table 2 shows that when the FIP was calculated for children with three doses of PCV13 in the Birindwa study, 93.2% (95% CI: 86.3–96.6) of *S. pneumoniae* carriage cases were prevented in children who received three doses of PCV13 compared with those who were not vaccinated. Considering the incidence of invasive pneumonia, the ORs reported by Coulibaly et al. show that the introduction of PCV13 prevented 66.7% (95% CI: 37.2–82.2) of cases of invasive pneumonia in vaccinated children compared with unvaccinated children. According to the measures of association obtained by Manegabe et al., the proportion of cases of meningitis attributable to *S. pneumoniae* prevented ranged from 6% to 93.3% in PCV13 children.

### 3.3. Overall Effect of PCV13 on the Burden of Pneumococcal Infections in the DRC

Figure 3A–C shows the effect of introducing PCV13 into the EPI schedule as a function of changes in vaccination coverage. The effect of PCV13 in the general population increased over time. This effect varied according to vaccination coverage. For example, when PCV13 vaccination coverage was 73.0%, the number of cases of severe pneumonia prevented was almost 100,000 per year (Figure 3A), while the number of deaths attributable to severe pneumonia was greater than 12,072 per year (Figure 3B). Thus, with PCV13 vaccination coverage at 79.0% in 2022, 113,359 new cases of severe pneumonia and 17,255 deaths attributable to severe pneumonia were prevented in the DRC (Figure 3C).

Figure 4A–C shows similar effects for pneumococcal meningitis. PCV13 vaccination coverage of 79.0% prevented more than 3000 new cases of pneumococcal meningitis (Figure 4A). Unlike pneumonia, where deaths were prevented even with low PCV13 coverage, for meningitis, only PCV13 coverage of over 60.0% prevented deaths in children suffering from meningitis (Figure 4B). Thus, in 2022, the number of new cases and deaths attributable to pneumococcal meningitis prevented in the DRC owing to PCV13 vaccination coverage at 79.0% were 3313 and 1544, respectively (Figure 4C).

We found no studies reporting *S. pneumoniae* serotypes in the DRC prior to the introduction of PCV13. Of all the studies included in this review (n = 4), only three reported data on *S. pneumoniae* serotypes.

The detailed distribution of *S. pneumoniae* serotypes identified in the included studies is presented in Table 3.

This table presents the results of three studies relating to *S. pneumoniae* serotypes. Birindwa’s study included healthy children who carried *S. pneumoniae* and the prevalence of carriage was 20.5% [26]. Serotype analysis showed that more than half of the serotypes identified were not among those currently covered by PCV13. Almost all the serotypes included in PCV13 were identified in PCV13 children. Of these, 19F was carried by 52.2% of children.

Serotypes not included in the vaccines were identified in 14.8% of vaccinated children (3 doses), while 85.2% were identified in non-PCV13 children. Among PCV13 children, 18 was carried by more than two-thirds (66.7%), while 19A was carried by 33.3%. In non-PCV13 children carrying serotypes not included in PCV13, 35B/35C was identified in almost a third of children (30.4%).

In children hospitalized for pneumonia [19], the distribution of serotypes according to their inclusion in PCV13 and the vaccination status of the children was similar to that observed in healthy carrier children. For example, we noted that in almost half of the children (47.4%), the serotypes identified were those included in PCV13. Serotypes 19F (44.4%), 5 (27.8%) and 19A (13.9%) were the most identified in children who had already been vaccinated with PCV13.

Concerning serotypes not included in PCV13, all were identified in non-PCV13 children. In non-PCV13 children, 35B/35C, 15B/C, 10A and 11A/D were the serotypes identified in most children.

In children with pneumococcal meningitis [27], only one serotype included in PCV13 was identified—19F among PCV13 children. As in the two previous studies, no serotypes included in PCV13 were identified in non-PCV13 children. On the other hand, most serotypes not included in PCV13 identified in PCV13 children were not the same as those identified in children suffering from pneumonia or carrying only *S. pneumoniae*. In the last category, 55.6% of isolated *S. pneumoniae* were of unknown serotypes.

## 4. Discussion

The studies analyzed in this review primarily examined the impact of PCV13 on *Streptococcus pneumoniae* among children under five. Their focus was on how the vaccine influences bacterial carriage, its role in preventing invasive pneumonia and pneumococcal meningitis, and the distribution of *S. pneumoniae* serotypes.

### 4.1. PCV13 Effects

Regarding carriage, it was noted that when children received three doses of PCV13, the prevalence of carriage decreased by 93.3%. A single dose of PCV13 did not significantly reduce the prevalence of carriage in PCV13-positive children compared with non-positive children. This level of effect is comparable to what has been observed in several African settings.

In Burkina Faso, for example, Kaboré et al. observed direct effects ranging from 12.2% to 72.0% [34]. In Laos, Chan J et al. observed that PCV13 coverage was associated with a reduction in the probability of PCV13 carriage. These authors noted a reduction of 38.1% (95%CI: 4.1% to 60.0%) in PCV13 children. For each percentage point increase in PCV13 coverage, the prevalence of carriage fell by 1.1% [35]. In the Republic of South Africa, the overall effect assessed by the reduction in the prevalence of pneumococcal colonization was 54.5% (aOR: 0.41; 95%CI: 0.3–0.56) [36].

In relation to the prevention of invasive pneumonia, this review found that PCV13 prevented 66.7% (95%CI: 37.2–82.2) of cases of invasive pneumonia in children. The only study to assess this effect did not specify the dose at which it was obtained.

This result is in line with those obtained by other researchers in low-income countries. For example, Bar-Zeev N et al. in Blantyre, Malawi, observed that a reduction in the total incidence (vaccine serotype plus non-vaccine serotype) of invasive pneumococcal disease followed PCV vaccine introduction: 19% in infants and 14% in children aged 1 to 4 years [37]. In the Gambia, Mackenzie GA noted that the reduction in the incidence of invasive pneumococcal disease was related to an 82% reduction in the serotypes covered by the PCV13 vaccine. In the 2–4-year age group, the incidence of invasive pneumococcal disease was reduced by 56%, with a 68% reduction in serotypes covered by PCV13. The incidence of serotypes other than PCV13 in children aged 2–59 months increased by 47%, with a wide range of serotypes [38].

Taking into account the prevalence of colonization with *S. pneumoniae* in the DRC, the incidence of pneumococcal pneumonia and meningitis, and PCV13 vaccination coverage among children under five years of age, it emerged that in terms of the effects of PCV13 on pneumococcal infections, in 2022, vaccination of 79.0% prevented 113,359 new cases of severe pneumonia and 17,255 pneumonia-related deaths, and 3313 new cases of meningitis and 1544 pneumococcal-meningitis-related deaths in children under five in the DRC.

### 4.2. PCV13 Serotypes

Several studies reported changes in *S. pneumoniae* serotypes following the introduction of PCV [5,6,33,39,40,41]. The available literature does not mention any study of *S. pneumoniae* serotypes in the DRC prior to PCV13 introduction. Thus, it was not possible to compare serotypes before and after PCV13 introduction. According to the studies included in this review, serotypes 19F and 23F, often carried by children under five, were also frequently responsible for invasive pneumonia in this age group [18,23,27].

As observed in South African infants, where non-PCV13 serotypes such as 15BC, 10A, 21, and 16F became the most prevalent after the introduction of PCV13 [10], in the DRC, serotypes 35B/35C, 15B/C, 10A, and 11A/D—although not included in PCV13—were among the most frequently identified causes of morbidity in children.

### 4.3. Limitations

The literature on pneumococcal vaccination in the DRC remains limited in volume, diversity, and geographical representativeness. This review revealed that 12 years after the introduction of PCV13 into the national immunization program, few scientific studies have been published on its impact on pneumococcal infection prevention or on *S. pneumoniae* serotype evolution. The three articles included in this review were limited in scope. Two of them [18,27] focused solely on urban areas of a single province, while the third, though comparing children from two major cities, was not representative of the broader urban or national population.

The effect reported by Birindwa in the DRC appears greater than that observed in other countries, which may be explained by the study’s small sample size. Moreover, the authors did not document the assumptions underlying their sample size calculation [26].

In Coulibaly et al.’s study [19], the effect of PCV13 was not adjusted for common confounding factors cited in the literature. The external validity of the findings is therefore uncertain both regarding the broader child population in the studied cities and those who became ill but did not attend the sentinel sites of Kalembelembe and Kingasani in Kinshasa or Sendwe in Lubumbashi. Additionally, because data on the number of PCV13 doses were not collected, it was impossible to assess a dose–response relationship. In the study by Manegabe et al. [27], statistical power was not considered when calculating sample size, which was instead determined by available funding.

Regarding analysis methods, vaccine coverage was treated as an estimate and assumed to be homogeneous, which it is not. Furthermore, many methods used are based on models incorporating numerous assumptions, such as the Indian model cited in this review.

In addition, the limited number of studies conducted in the DRC on this subject influenced the sample size used for association measures and, consequently, the overall estimate of vaccine impact. This estimate could vary as more studies are conducted, more subjects are included, and vaccine coverage approaches the herd immunity threshold.

However, despite these methodological limitations, the available studies provide valuable insights that enabled estimation of the vaccine’s effect in the population based on observed data. The scarcity of robust studies on this vaccine’s effects highlights the need to conduct further research that reflects the DRC’s geographical and demographic diversity, as well as the heterogeneity of vaccination coverage.

## 5. Conclusions

We identified major gaps in the existing literature, including the lack of studies on the effect of PCV13 conducted in settings with the highest under-five mortality, and the lack of studies that reflect the geographic and socio-behavioral diversity of the Democratic Republic of Congo. There is a need for large-scale studies in contexts of high morbidity and mortality to simultaneously demonstrate the effect of the vaccine on both disease occurrence and related deaths.

There is clear, but scattered, evidence of reduced *Streptococcus pneumoniae* colonization and hospital admissions due to radiologically confirmed pneumonia and pneumococcal meningitis. Evidence has also shown that *S. pneumoniae* serotypes 35B/35C, 15B/C, 10A and 11A/D, which are not covered by PCV13, contribute to pneumococcal disease. These data support the need to improve vaccination coverage among children with low or zero coverage to reduce the burden of pneumococcal infections, and to establish and strengthen sentinel surveillance of invasive bacterial diseases in the Democratic Republic of Congo.

## Figures and Tables

**Figure 1 vaccines-13-00603-f001:**
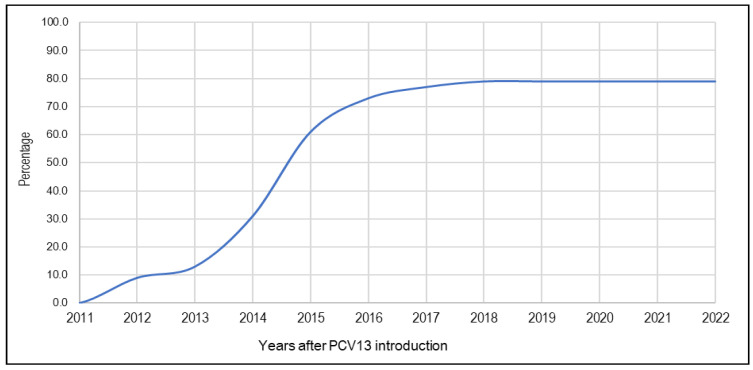
Trends in vaccination coverage against pneumococcal infections since the introduction of PCV13.

**Figure 2 vaccines-13-00603-f002:**
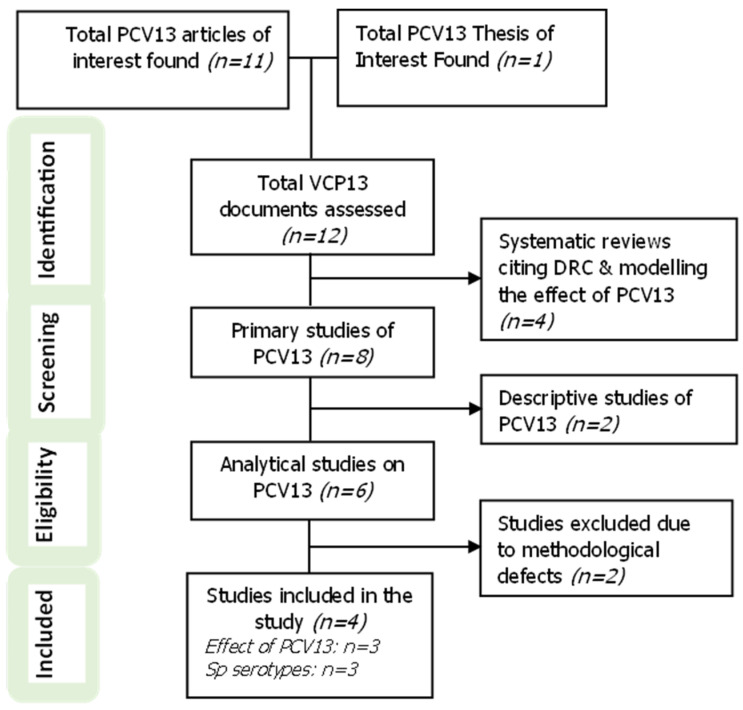
Inclusion of studies in this systematic review (PRISMA flow diagram of study selection).

**Figure 3 vaccines-13-00603-f003:**
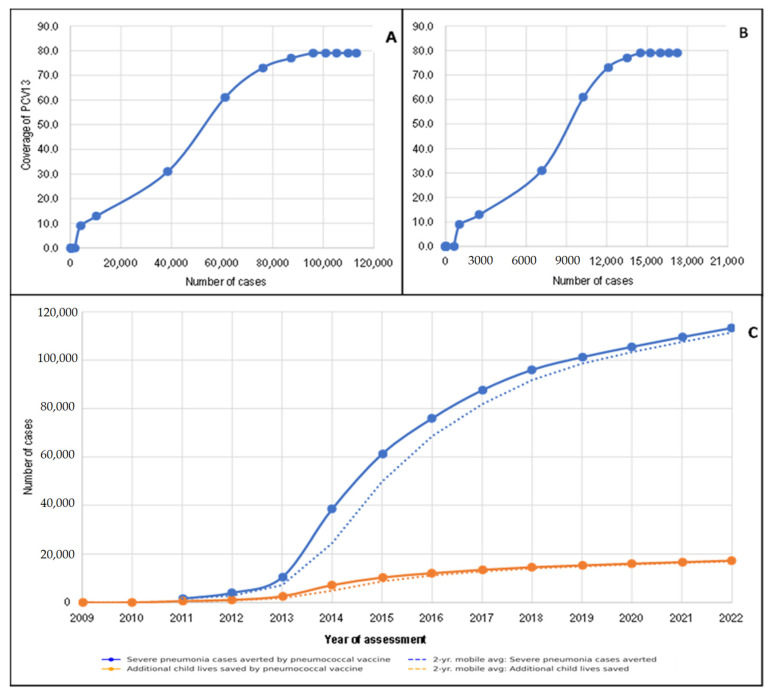
Variation in the number of cases of severe pneumonia preventable by PCV13 (**A**) and cases of death due to *S. pneumoniae* pneumonia preventable by PCV13 (**B**) according to vaccination coverage and year (**C**).

**Figure 4 vaccines-13-00603-f004:**
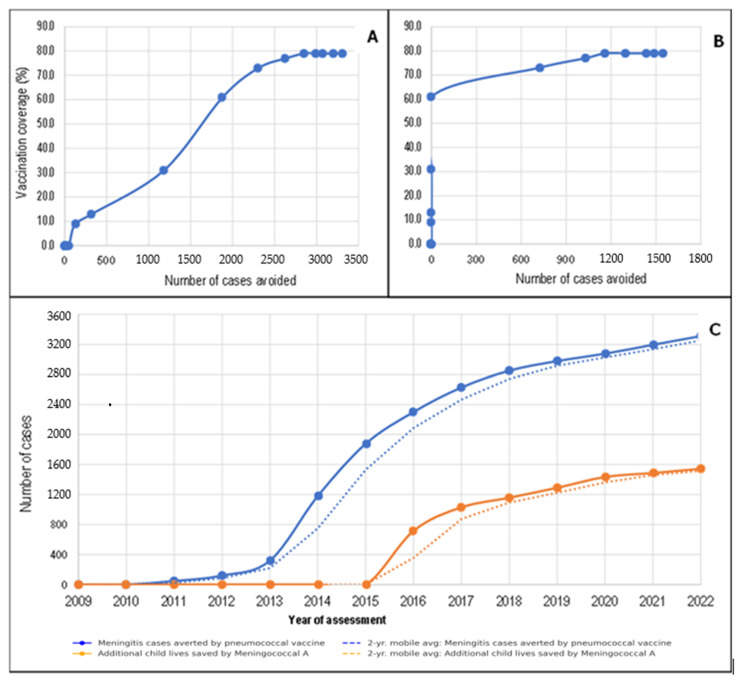
Variation in the number of cases of meningitis (**A**) and deaths due to *S. pneumoniae* meningitis preventable by PCV13 (**B**) according to vaccination coverage and year (**C**).

**Table 1 vaccines-13-00603-t001:** Characteristics of the studies included in this systematic review.

Ref.	Author, Year	Specific Objective	Medium	Children’s Profile	No.	Type of Study	Study Period (Data Collection)	Judgment Criteria	Comparison	Measurement of Effect
[26]	Birindwa, A.M., 2018	Nasopharyngeal carriage and conditions predisposing to pneumococcal colonization in healthy children after PCV13 introduction	South Kivu: Health Centers during preschool consultations	Healthy children aged 1 to 60 months	794	Cross-sectional	2014 to June 2015	*S. pneumoniae* carriage	PCV13 vs. Non-PCV13	2–3 vs. 0 doses: aOR = 0.07 95% CI: 0.03–0.14 1 vs. 0 dose: aOR = 0.84 95% CI: 0.55–1.27
[19]	Coulibaly, A., 2016	To examine the relationship between childhood PCV13 immunization rates in DRC provinces and the incidence of invasive pneumococcal disease	Kinshasa: sentinel surveillance sites of Kalembelembe and Kingasani, and Lubumbashi at Sendwe hospital	Sick children treated in sentinel surveillance sites	380	Quasi-experimental, with control group	2009, July 2013	Incidence of invasive pneumococcal disease	PCV13 vs. Non-PCV13	aOR = 0.33 95% CI: 0.63–0.18
[27]	Manegabe, J.T., 2023	Determining the presence of bacteria and viruses in the CSF of hospitalized children with meningitis	South Kivu: pediatric department of Panzi hospital	Children hospitalized due to meningitis	150	Cross-sectional	April 2021 to end of March 2022	CSF Presence of *S. Pneumoniae* in children	PCV13 vs. non-PCV13	OR = 4.0 95% CI: 1.06–15.0

**Table 2 vaccines-13-00603-t002:** Fraction of infections prevented in children vaccinated with PCV13.

Author, Year	Outcomes	Number of Children	Odds Ratio (95%CI)	FIP at PCV13 % (95%CI)
Birindwa, A.M., 2018 [26]	*S. pneumoniae* carriage	794	0.07 (0.03–0.14) †	93.2 (86.3–96.6)
Coulibaly, A., 2016 [19]	Incidence of invasive pneumococcal disease	380	0.33 (0.63–0.18)	66.7 (37.2–82.2)
Manegabe, J.T., 2023 [27]	Presence of *S. pneumoniae* in children	150	4.0 (1.06–15.0)	75.0 (6.0–93.3)

FIP: Fraction of infections prevented (%); †: 2–3 vs. 0 dose.

**Table 3 vaccines-13-00603-t003:** *S. pneumoniae* serotypes identified in DRC studies post PCV13 vaccine introduction.

Children’s Profile	Included in PCV13	Birindwa, A.M., 2018 [26]	Birindwa, A.M., 2020 [18]	Manegabe, J.T., 2023 [27]
		Healthy Children	2 to 59 Months Hospitalized for Pneumonia	Children Hospitalized with Meningitis
Children with isolated serotypes(n)		163	375	37
*% with S. pneumoniae*		*20.5*	*77.0*	*37.8*
**Included in PCV13** (%) ⱡ		**46.0**	**47.4**	**10.2**
Children PCV13 (%) ꬸ		100.0	100.0	*100.0*
*1*	*Yes*	*0.0*	*2.8*	*0.0*
*19A*	*Yes*	8.7	*13.9*	*0.0*
*19F*	*Yes*	*52.2*	*44.4*	*100.0*
*14*	*Yes*	*4.3*	*2.8*	*0.0*
*23F*	*Yes*	*8.7*	*2.8*	*0.0*
*9A/V*	*Yes*	*4.3*	*5.6*	*0.0*
*3*	*Yes*	*8.7*		*0.0*
*6ABCD*	*Yes*	*8.7*		*0.0*
*5*	*Yes*	*4.3*	*27.8*	*0.0*
Non-PCV13 children (%)		*0.0*	*0.0*	*0.0*
*18C*	*Yes*	*0.0*	*0.0*	*0.0*
*4*	*Yes*	*0.0*	*0.0*	*0.0*
**Not included in PCV13**		**54.0**	**52.6**	**89.8**
Child PCV13 (%)		14.8	0.0	73.3
*Unknown*		*0.0*	*0.0*	*55.6*
*13*		*0.0*	*0.0*	*11.1*
*18*		100.0	*0.0*	*0.0*
*23A*		*0.0*	*0.0*	*2.0*
*22F*		*0.0*	*0.0*	*11.1*
*33F*		*0.0*	*0.0*	*11.1*
*15B*		*0.0*	*0.0*	*11.1*
Non-PCV13 children (%)		85.2	100.0	26.7
*11A/D*		*21.7*	*15.0*	*0.0*
*10A*		*8.7*	*15.0*	*0.0*
*15B/C*		*4.3*	*17.5*	*0.0*
*12*		*4.3*	*7.5*	*0.0*
*35B/35C*		*30.4*	*17.5*	*0.0*
*20*		*0.0*	*2.5*	*0.0*
*38*		*0.0*	*7.5*	*0.0*
*34/17A*		*13.0*	*0.0*	*0.0*
*9N/L*		*4.3*	*0.0*	*0.0*
*17*		*4.3*	*0.0*	*0.0*
*2*		*4.3*	*2.5*	*66.7*
*7C*		*4.3*	*7.5*	*33.3*
*22F*		*0.0*	*2.5*	*0.0*
*20*		*0.0*	*2.5*	*0.0*
*7AF*		*0.0*	*2.5*	*0.0*

ⱡ: calculated in relation to the total number of children in whom the serotype was identified; ꬸ: calculated in relation to the total number of children carrying the serotypes included or not in PCV13.

## Data Availability

The data presented in this study are available on request from the WHO-DRC office at the email address “nimpamengouom@who.int”.

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
