# Peer review of "Effects of Pneumococcal Vaccination in Children Under Five Years of Age in the Democratic Republic of Congo: A Systematic Review"

_vaccines, 2025, doi:10.3390/vaccines13060603_

Round 1
Reviewer 1 Report
Comments and Suggestions for Authors
I have finished the review of “Effects of pneumococcal vaccination in children under five years of age in the Democratic Republic of Congo: a systematic review” finding that this manuscript investigates the vaccination coverage and transmission of pneumococcal infections since 2011.
For adding originality and relevance to the Field is very limited, because by making a systematic review, a more comprehensive area could be addressed
Contribution to the Subject Area Compared with other published materials. It is limited because of its design, public registries of surveillance are included along with thesis and research articles, that makes a structure that does not ensure that datum are not repeatedly accounted.
Methodological Improvements are needed. It is not correct to estimate prevalence ratios for cases of pneumococcal infections, because for a prevalence, the research unit must be a sample of individuals representative of a population, enough as estimated with sample size calculation and random, even more, the disease can only happen once, because prevalence is a frequency measure for estimating long term diseases. Incidences are used as frequency measures for acute illnesses; for estimating incidence rates, a denominator must be adjusted for population size.
Search Strategy: Provide the link(s) to PubMed and other databases search. It should be reproductible to be rigorous enough for publication.
Is the protocol registered? Where is it registered?
Elaborate on how statistical methods were used to analyze the data, how did you manage to avoid including repeated individuals (more than one infection) in a same patient and estimate prevalence?
Discuss how confounding variables were accounted for during data analysis.
Mention what you did to reduce selection bias, such as outlining efforts to ensure the systematic inclusion of diverse case reports.
Consistence of Conclusions with Evidence. The authors should explicitly acknowledge these limitations in the discussion.
Appropriateness of References The references are largely appropriate and relevant to the topic. Why thesis? Why not only include peer-reviewed articles?
The tables are informative but could benefit from additional clarity not even the font is uniform. Figures: Ensure that all figures are of high resolution and adequately labeled. outcomes.
Please correct the use of English. Professional editing is recommended to ensure clarity.

English editing is reccomended, some paragraphs are difficult to understand.
Author Response
Rev 1
Comments and Suggestions for Authors
I have finished the review of “Effects of pneumococcal vaccination in children under five years of age in the Democratic Republic of Congo: a systematic review” finding that this manuscript investigates the vaccination coverage and transmission of pneumococcal infections since 2011.
For adding originality and relevance to the Field is very limited, because by making a systematic review, a more comprehensive area could be addressed
Contribution to the Subject Area Compared with other published materials. It is limited because of its design, public registries of surveillance are included along with thesis and research articles, that makes a structure that does not ensure that datum are not repeatedly accounted.
Methodological Improvements are needed. It is not correct to estimate prevalence ratios for cases of pneumococcal infections, because for a prevalence, the research unit must be a sample of individuals representative of a population, enough as estimated with sample size calculation and random, even more, the disease can only happen once, because prevalence is a frequency measure for estimating long term diseases. Incidences are used as frequency measures for acute illnesses; for estimating incidence rates, a denominator must be adjusted for population size.
Response: In some cases, the term prevalence was used in the original studies included in this review. In some cases, babywearing is equivalent to prevalence. Finally, for infections with a short course as is the case for this study, the incidence values are equal to those of the incidence, often considered as an attack rate. For the denominator, the studies used the population likely to develop pneumonia or pneumococcal meningitis, however the adjustment was not made for time as it is a cumulative measure.
Search Strategy: Provide the link(s) to PubMed and other databases search. It should be reproductible to be rigorous enough for publication.
Response: We included the bibliographic search strategy, with keywords and their combination, to allow other researchers to find the documents included in this systematic review (see lines 144-148)
Is the protocol registered? Where is it registered?
Response: This protocol has been registered; See lines 118-120
The systematic review protocol was registered in the International Prospective Registry of Systematic Reviews (PROSPERO), with registration number CRD42025641327 (https://www.crd.york.ac.uk/prospero/#recordDetails)
Elaborate on how statistical methods were used to analyze the data, how did you manage to avoid including repeated individuals (more than one infection) in a same patient and estimate prevalence?
Response: We did not use the primary databases of the studies included in this systematic review. We extracted aggregate data and calculated useful epidemiological measures. In primary studies, the authors did not analyze the occurrence of these infections in the form of episodes; implying that everyone was included only once in the study.
Discuss how confounding variables were accounted for during data analysis.
Response: We consider the Herd effect in the model as the additional proportion of cases or deaths due to vaccine-susceptible pneumococci that could be prevented by population-level PCV coverage, in addition to the cases and deaths prevented directly by vaccination.
This paragraph has been added in the manuscript (see lines 225-227)
Mention what you did to reduce selection bias, such as outlining efforts to ensure the systematic inclusion of diverse case reports.
Response: The paragraph below has been added in the manuscript (see lines 157-162)
After having carried out all the bibliographic research on all published and unpublished documents on the DRC or in Africa, reporting data on the impact of PCV13 vaccination on the incidence of pneumonia and pneumococcal meningitis, we organized working meetings with experts in the field of surveillance of the germs concerned, to search for additional reports ―to ensure the completeness of the reports collected― that could be taken into account in this systematic review.
Consistence of Conclusions with Evidence. The authors should explicitly acknowledge these limitations in the discussion.
Response: The paragraph below has been added in the manuscript (see lines 409-413)
In addition, the limited number of studies in the DRC on the subject addressed in this study influenced the number of subjects used for the calculation of association measures and consequently the overall impact of vaccination. This impact could change in one direction or another as the number of studies and subjects included in this evaluation increases, and vaccination coverage in the population reaches the threshold of herd effect.
Appropriateness of References The references are largely appropriate and relevant to the topic. Why thesis? Why not only include peer-reviewed articles?
Response: We have included this thesis because the various articles that make it up were in preparation for publication. We subjected these manuscripts to a critical reading grid and observed that these manuscripts were acceptable for publication. Therefore, we decided to include them.
The tables are informative but could benefit from additional clarity not even the font is uniform.
Response: This could be formatted with the review formatting process
Figures: Ensure that all figures are of high resolution and adequately labeled. outcomes.
Response: Done
Please correct the use of English. Professional editing is recommended to ensure clarity.
Comments on the Quality of English Language
English editing is recommended, some paragraphs are difficult to understand.
Response: Done
Reviewer 2 Report
Comments and Suggestions for Authors
Summary
The authors assert that PCV13 was introduced into the DRC in 2011 in a three dose regimen for infants as part of the country’s EPI programme. The aim of their study was to determine the proportion of pneumonia and meningitis cases and deaths in children < 5y/o by this vaccine.
They conducted a systematic review of research conducted in the DRC from 2011 to 2023 of published and non-published materials, and selected only those reporting results from original analyses in French or English were selected. They assessed the direct effect of PCV13 by calculating the fraction of infections avoided.
They report the results from four studies. They found, amongst children receiving three doses, that carriage was reduced by 93.3%, and an insignificant reduction in children receiving only one dose compared to non-vaccinated children. Three doses prevented pneumonia by 66.7% and meningitis by 75.0% compared to non-vaccinated children. Serotypes 19F and 23F were the most frequent causes of invasive pneumonia in children, while 35B/35C, 15B/C, 10A and 11A/D were the most frequent causes of morbidity. They report that in 2022, with vaccine coverage at 79.0%, 113,359 new cases of severe pneumonia and 17,255 deaths attributable to the same were prevented; whilst 3,313 cases and 1,544 deaths attributable to meningitis were prevented.
They conclude there there is evidence of reduced colonisation by S pneumoniae, and hospital admissions associated with pneumococcal pneumonia and meningitis. They also conclude that the serotypes above that are not included in PCV13 were the main causes of pneumococcal disease in non-vaccinated children, and that their data indicate the need for continued improvement in vaccination coverage.
General comments
The introduction is thorough, well-referenced with mostly up to date references for specific points, which combined provide a rationale for the present study.
In the methods section, the authors provide the details of their use of the PRISMA guidelines, and their registration with PROSPERO. They provide demographic information regarding the DRC for 2022, but do not provide a source, as they do with their 2009 data. This should be corrected (eg CIA Fact Book, national records, etc). The authors cite their methodology for figure 1 which provides vaccination coverage trends for the period of analysis. The authors’ search strategy is appropriately documented, including databases, search terms (keywords), and the use of the Effective Public Health Practice Project’s tool for assessing the quality of identified studies. Their inclusion criteria are clearly described. They provide a description of their analysis plan, their use of data from meta-analyses vs original studies, and a rationale for not conducting a meta-analysis. They further provide the formula by which they assessed burden of pneumococcal infection. All of these appear to be sufficiently descriptive and detailed to appreciate their methodology.
In their results, figure 2 provides a PRISMA flow diagram that corresponds to their data, and a table showing the characteristics of the included studies. What surfaces is that only four studies met their inclusion criteria. Though the data set is limited, it has the advantage of spanning the gamut of their reporting period, however two of the studies cover the same province, raising questions about generalisability. Two of the studies were cross-sectional in design, and one used quasi-experimental prospective data comparing pneumococcal infections between a vaccinated vs a non-vaccinated province. Of note, all three of the included studies were relatively small, with a combined study population of 1,324 children. The calculations derived from these data appear to be appropriate. What lends confidence to their findings, despite limited data, is that the effect of PCV13 varied with the coverage, with a greater effect with greater coverage. These are depicted in their figure 3A-C for pneumonia, and figures 4A-C for meningitis. Despite the relatively small sample size of their review, table 1 does demonstrate the impact of vaccination on carriage.
A not-surprising observation is that the authors found no studies reporting pneumococcal serotypes prior to PCV13 introduction, and only three of four studies reported serotype carriage by their report. It is difficult to make much of the carriage findings, given the limited data. One observation is the relatively low carriage of 20.5% amongst healthy children. Also noteworthy is that, amongst the pneumonia cases, 42.3% of the identified serotypes were included in PCV13, while amongst meningitis cases, only two of the PCV13 serotypes were identified. This reporting of the results would be clearer and easier to assess epidemiologically if presented in terms of burden than numbers. For example, of the meningitis cases amongst vaccinated vs non-vaccinated children, what proportion were vaccine vs non-vaccine serotypes, and similarly for pneumonia. With such small numbers, while one cannot make too much of this breakdown, it would be more meaningful than a count of the recovered serotypes. The authors can comment.
In their discussion, the authors’ use of the South African experience with PCV13 and the emergence of non-vaccine serotypes highlighted by Fleming-Dutra et al [PIDJ. 2014] is both sound and plausible. In the absence of pre-introduction data, the only remaining option is further surveillance data from the DRC amongst vaccinated and non-vaccinated districts and children.
In the section, the authors are likely correct that the effect reported in one of the DRC publications is likely due to the sample size. Indeed, all of the findings in this analysis are a case study of brilliant but ultimately limited capacity to extrapolate. The study limitations appear otherwise complete. The conclusion is concise and comprehensive.
Ultimately, more data are needed from this setting. Nonetheless, this paper makes an important contribution to the scientific community from an under-represented but important region.
Specific comments
- The statistics attributed to WHO on page 2, lines 53-54, should be referenced, even if it only an online source.
- If citations 13 and 14 have either a url, DOI or ISSN identifier, these should be added.
- Page 13, line 351; the formatting of 79.0% should be revised for clarity (shows 79,0%)
Author Response
Rev 2
Comments and Suggestions for Authors
Summary
The authors assert that PCV13 was introduced into the DRC in 2011 in a three-dose regimen for infants as part of the country’s EPI programme. The aim of their study was to determine the proportion of pneumonia and meningitis cases and deaths in children < 5y/o by this vaccine.
They conducted a systematic review of research conducted in the DRC from 2011 to 2023 of published and non-published materials and selected only those reporting results from original analyses in French or English were selected. They assessed the direct effect of PCV13 by calculating the fraction of infections avoided.
They report on the results from four studies. They found, amongst children receiving three doses, that carriage was reduced by 93.3%, and an insignificant reduction in children receiving only one dose compared to non-vaccinated children. Three doses prevented pneumonia by 66.7% and meningitis by 75.0% compared to non-vaccinated children. Serotypes 19F and 23F were the most frequent causes of invasive pneumonia in children, while 35B/35C, 15B/C, 10A and 11A/D were the most frequent causes of morbidity. They report that in 2022, with vaccine coverage at 79.0%, 113,359 new cases of severe pneumonia and 17,255 deaths attributable to the same were prevented; whilst 3,313 cases and 1,544 deaths attributable to meningitis were prevented.
They conclude there is evidence of reduced colonisation by S pneumoniae, and hospital admissions associated with pneumococcal pneumonia and meningitis. They also conclude that the serotypes above that are not included in PCV13 were the main causes of pneumococcal disease in non-vaccinated children, and that their data indicate the need for continued improvement in vaccination coverage.
General comments
The introduction is thorough, well-referenced with mostly up to date references for specific points, which combined provide a rationale for the present study.
In the methods section, the authors provide details of their use of the PRISMA guidelines, and their registration with PROSPERO. They provide demographic information regarding the DRC for 2022, but do not provide a source, as they do with their 2009 data. This should be corrected (eg CIA Fact Book, national records, etc). The authors cite their methodology for figure 1 which provides vaccination coverage trends for the period of analysis. The authors’ search strategy is appropriately documented, including databases, search terms (keywords), and the use of the Effective Public Health Practice Project’s tool for assessing the quality of identified studies. Their inclusion criteria are clearly described. They provide a description of their analysis plan, their use of data from meta-analyses vs original studies, and a rationale for not conducting a meta-analysis. They further provide the formula by which they assessed burden of pneumococcal infection. All of these appear to be sufficiently descriptive and detailed to appreciate their methodology.
In their results, figure 2 provides a PRISMA flow diagram that corresponds to their data, and a table showing the characteristics of the included studies. What surfaces is that only four studies met their inclusion criteria. Though the data set is limited, it has the advantage of spanning the gamut of their reporting period, however two of the studies cover the same province, raising questions about generalisability. Two of the studies were cross-sectional in design, and one used quasi-experimental prospective data comparing pneumococcal infections between a vaccinated vs a non-vaccinated province. Of note, all three of the included studies were relatively small, with a combined study population of 1,324 children. The calculations derived from these data appear to be appropriate. What lends confidence to their findings, despite limited data, is that the effect of PCV13 varied with the coverage, with a greater effect with greater coverage. These are depicted in their figure 3A-C for pneumonia, and figures 4A-C for meningitis. Despite the relatively small sample size of their review, table 1 does demonstrate the impact of vaccination on carriage.
Response: We discussed within the limitations of the study the implication of this limited number of studies and sample size; including the role of the Herd effect.
A not-surprising observation is that the authors found no studies reporting pneumococcal serotypes prior to PCV13 introduction, and only three of four studies reported serotype carriage by their report. It is difficult to make much of the carriage findings, given the limited data. One observation is the relatively low carriage of 20.5% amongst healthy children. Also noteworthy is that, amongst the pneumonia cases, 42.3% of the identified serotypes were included in PCV13, while amongst meningitis cases, only two of the PCV13 serotypes were identified. This reporting of the results would be clearer and easier to assess epidemiologically if presented in terms of burden than numbers. For example, of the meningitis cases amongst vaccinated vs non-vaccinated children, what proportion were vaccine vs non-vaccine serotypes, and similarly for pneumonia. With such small numbers, while one cannot make too much of this breakdown, it would be more meaningful than a count of the recovered serotypes. The authors can comment.
Response: The numbers were so small for each serotype that it was difficult to achieve more subdivisions.
In their discussion, the authors’ use of the South African experience with PCV13 and the emergence of non-vaccine serotypes highlighted by Fleming-Dutra et al [PIDJ. 2014] is both sound and plausible. In the absence of pre-introduction data, the only remaining option is further surveillance data from the DRC amongst vaccinated and non-vaccinated districts and children.
Response: This suggestion was added to the manuscript
In the section, the authors are likely correct that the effect reported in one of the DRC publications is likely due to the sample size. Indeed, all of the findings in this analysis are a case study of brilliant but ultimately limited capacity to extrapolate. The study limitations appear otherwise complete. The conclusion is concise and comprehensive.
Ultimately, more data are needed from this setting. Nonetheless, this paper makes an important contribution to the scientific community from an under-represented but important region.
Specific comments
- The statistics attributed to WHO on page 2, lines 53-54, should be referenced, even if it only an online source.
- If citations 13 and 14 have either a url, DOI or ISSN identifier, these should be added.
==Done
- Page 13, line 351; the formatting of 79.0% should be revised for clarity (shows 79,0%)
Responses: ===Done
Reviewer 3 Report
Comments and Suggestions for Authors
This research is interesting but the manuscript needs to be improved.
Throughout the article the scientific writing of the name of the microorganism (Streptococcus pneumoniae, S. pneumoniae) does not comply with the adopted rules – you should italicize the names in the text.
Introduction
Lines 68-73: In the text, the authors write that there are two types of vaccines but only explain PCV13, perhaps because it was the vaccine included in the Expanded Program on Immunization. It seems to me that you should complete the text regarding the type of vaccines that exist.
Lines 85-87: For me, the authors should indicate the recommended vaccination schedule and indicate the backup vaccination schedule when the first one is not followed.
Results
3.2. Profile of studies included in the review and effects of PCV13
Lines 226-234: The date the studies were carried out is different from the date of publication, so the authors must correct what they wrote in the text.
Lines 244-250: The risk of disease (i.e. the probability of the disease developing) cannot always be interpreted in the same way as the possibility (odds) of the disease developing. It seems to me that you are interpreting these concepts in the same way. Can you explain? I would also like you to explain the following statements: “the risk of S. pneumoniae carriage was 6 times lower in PCV13”, “the incidence of invasive pneumonia was 5 times higher in non-PCV13”.
Table 2: Where it says “FIP in PCV13%” it should say “FIP in PCV13 (%)”
3.3. Overall effect of PCV13 on the burden of pneumococcal infections in the DRC
Line 265-266: “when PCV13 vaccination coverage was 73.0%, the number of cases of severe pneumonia prevented was greater than 100,000 per year” – not what the figure shows.
Figure 3A and 3B: It makes more sense for the x-axis to correspond to vaccination coverage, and the y-axis to correspond to the number of cases prevented. Authors should clearly indicate that figure 3A refers to cases of disease and that figure 3B refers to deaths.
Figure 3C: The caption is very difficult to read, even when enlarging the figure.
Lines 271-273: Replace “PCV3” with “PCV13”. Keep the Arabic numerals used in the text.
Figure 4: Comments identical to those in Figure 3.
Table 3: In the first column, serotype 9B is indicated/analysed as included in PCV13, but this serotype is not included in this vaccine. Serotype 19A is indicated/analysed as not included in PCV13, but this serotype is included in the vaccine.
Line 296: The spelling should be changed to avoid misleading – what the authors mean is that “serotypes not included in the vaccines were identified in 20.7% of vaccinated children”.
Line 301: There is an error in the reference number indicated: the reference indicated in Table 3. is reference [18].
Line 304-305: “Serotype 5 was identified in almost two-thirds of PCV13 children” – the value indicated in the table is very different from two-thirds.
Lines 311-313: What the authors describe does not match what they present in the table.
References
Some references are not correct/complete, so the entire list should be revised.
Some examples:
Reference [19], presented in the list of references, does not correspond to the article included in this systematic review (only the author's name and the year of publication are correct).
In reference [28] the name of the scientific journal was not written.
Reference [21] is quite incomplete.
Comments on the Quality of English LanguageThis research is interesting but the manuscript needs to be improved.
Throughout the article the scientific writing of the name of the microorganism (Streptococcus pneumoniae, S. pneumoniae) does not comply with the adopted rules – you should italicize the names in the text.
Introduction
Lines 68-73: In the text, the authors write that there are two types of vaccines but only explain PCV13, perhaps because it was the vaccine included in the Expanded Program on Immunization. It seems to me that you should complete the text regarding the type of vaccines that exist.
Lines 85-87: For me, the authors should indicate the recommended vaccination schedule and indicate the backup vaccination schedule when the first one is not followed.
Results
3.2. Profile of studies included in the review and effects of PCV13
Lines 226-234: The date the studies were carried out is different from the date of publication, so the authors must correct what they wrote in the text.
Lines 244-250: The risk of disease (i.e. the probability of the disease developing) cannot always be interpreted in the same way as the possibility (odds) of the disease developing. It seems to me that you are interpreting these concepts in the same way. Can you explain? I would also like you to explain the following statements: “the risk of S. pneumoniae carriage was 6 times lower in PCV13”, “the incidence of invasive pneumonia was 5 times higher in non-PCV13”.
Table 2: Where it says “FIP in PCV13%” it should say “FIP in PCV13 (%)”
3.3. Overall effect of PCV13 on the burden of pneumococcal infections in the DRC
Line 265-266: “when PCV13 vaccination coverage was 73.0%, the number of cases of severe pneumonia prevented was greater than 100,000 per year” – not what the figure shows.
Figure 3A and 3B: It makes more sense for the x-axis to correspond to vaccination coverage, and the y-axis to correspond to the number of cases prevented. Authors should clearly indicate that figure 3A refers to cases of disease and that figure 3B refers to deaths.
Figure 3C: The caption is very difficult to read, even when enlarging the figure.
Lines 271-273: Replace “PCV3” with “PCV13”. Keep the Arabic numerals used in the text.
Figure 4: Comments identical to those in Figure 3.
Table 3: In the first column, serotype 9B is indicated/analysed as included in PCV13, but this serotype is not included in this vaccine. Serotype 19A is indicated/analysed as not included in PCV13, but this serotype is included in the vaccine.
Line 296: The spelling should be changed to avoid misleading – what the authors mean is that “serotypes not included in the vaccines were identified in 20.7% of vaccinated children”.
Line 301: There is an error in the reference number indicated: the reference indicated in Table 3. is reference [18].
Line 304-305: “Serotype 5 was identified in almost two-thirds of PCV13 children” – the value indicated in the table is very different from two-thirds.
Lines 311-313: What the authors describe does not match what they present in the table.
References
Some references are not correct/complete, so the entire list should be revised.
Some examples:
Reference [19], presented in the list of references, does not correspond to the article included in this systematic review (only the author's name is correct).
In reference [28] the name of the scientific journal was not written.
Reference [21] is quite incomplete.
Author Response
Rev 3
Comments and Suggestions for Authors
This research is interesting, but the manuscript needs to be improved.
Throughout the article the scientific writing of the name of the microorganism (Streptococcus pneumoniae, S. pneumoniae) does not comply with the adopted rules – you should italicize the names in the text.
Response: ===Done
Introduction
Lines 68-73: In the text, the authors write that there are two types of vaccines but only explain PCV13, perhaps because it was the vaccine included in the Expanded Program on Immunization. It seems to me that you should complete the text regarding the type of vaccines that exist.
Response: The sentence below has been completed in the text:
The second vaccine is unconjugated and contains 23 pneumococcal serotypes (1, 2, 3, 4, 5, 6B, 7F, 8, 9N, 9V, 10A, 11A, 12F, 14, 15B, 17F, 18C, 19A, 19F, 20, 22F, 23F and 33F). Although this vaccine has a broader spectrum, it is not effective before the age of two, and is not capable of eliminating pneumococcal carriage in the throat― the source of person-to-person transmission ―the protection conferred is short-lived, and there is no booster effect in the event of a new vaccine injection[9].
Lines 85-87: For me, the authors should indicate the recommended vaccination schedule and indicate the backup vaccination schedule when the first one is not followed.
Response: The sentence below has been completed in the text
« However, children can also be vaccinated at any contact for catch-up, when the usual vaccination schedule has not been respected”.
Results
3.2. Profile of studies included in the review and effects of PCV13
Lines 226-234: The date the studies were carried out is different from the date of publication, so the authors must correct what they wrote in the text.
Response: Precision has been given in the table
Lines 244-250: The risk of disease (i.e. the probability of the disease developing) cannot always be interpreted in the same way as the possibility (odds) of the disease developing. It seems to me that you are interpreting these concepts in the same way. Can you explain? I would also like you to explain the following statements: “the risk of S. pneumoniae carriage was 6 times lower in PCV13”, “the incidence of invasive pneumonia was 5 times higher in non-PCV13”.
Response: For the first two studies in Table 2, the type of study used allowed the production of measures that could be assimilated to cumulative incidence, thus enabling the calculation of relative risk. However, as multivariable models cannot generate RRs, AORs were used and interpreted as RRs.
Modification made for the interpretation of the results of study 3 (Manegabe JT, 2023): see lines 253-254.
Table 2: Where it says “FIP in PCV13%” it should say “FIP in PCV13 (%)”
Response: Has been Corrected
3.3. Overall effect of PCV13 on the burden of pneumococcal infections in the DRC
Line 265-266: “when PCV13 vaccination coverage was 73.0%, the number of cases of severe pneumonia prevented was greater than 100,000 per year” – not what the figure shows.
Response: Has been Corrected
Figure 3A and 3B: It makes more sense for the x-axis to correspond to vaccination coverage, and the y-axis to correspond to the number of cases prevented. Authors should clearly indicate that figure 3A refers to cases of disease and that figure 3B refers to deaths.
Response: Has been Corrected
Figure 3C: The caption is very difficult to read, even when enlarging the figure.
Response: Correction to be made during formatting
Lines 271-273: Replace “PCV3” with “PCV13”. Keep the Arabic numerals used in the text.
Response: Has been Corrected
Figure 4: Comments identical to those in Figure 3.
Response: Has been Corrected
Table 3: In the first column, serotype 9B is indicated/analysed as included in PCV13, but this serotype is not included in this vaccine. Serotype 19A is indicated/analysed as not included in PCV13, but this serotype is included in the vaccine.
Response: Table 2 has been corrected
Line 296: The spelling should be changed to avoid misleading – what the authors mean is that “serotypes not included in the vaccines were identified in 20.7% of vaccinated children”.
Response: This modification has been added in the manuscript
Line 301: There is an error in the reference number indicated: the reference indicated in Table 3. is reference [18].
Line 304-305: “Serotype 5 was identified in almost two-thirds of PCV13 children” – the value indicated in the table is very different from two-thirds.
Response: Modification has been added in the manuscript
Lines 311-313: What the authors describe does not match what they present in the table.
Response: Modification has been added in the manuscript
References
Some references are not correct/complete, so the entire list should be revised.
Some examples:
Reference [19], presented in the list of references, does not correspond to the article included in this systematic review (only the author's name and the year of publication are correct).
Response: Corrected
In reference [28], the name of the scientific journal was not written.
Response: Corrected
Reference [21] is quite incomplete.
Response: Corrected
Round 2
Reviewer 1 Report
Comments and Suggestions for Authors
The manuscript has improved.
Nevertheless, the complete text between 181-191 corresponding to the data processing and statistical analysis is completely identical to the same section from an already published paper: https://journals.plos.org/plosone/article?id=10.1371%2Fjournal.pone.0203186#sec006 in which the following text can be found:
"Data processing and statistical analysis
Data were entered into an Epi Info software program, version 3.5.4, exported to Microsoft Excel, and analyzed using Statistical Package for the Social Sciences (SPSS) version 20, IBM. Descriptive statistics were used to summarize the characteristics of the study population. Continuous variables were reported as follows: the mean with standard deviation for patient’s age, and the median with interquartile range for length of hospital stay.
The Mann-Whitney test was used to compare the median length of hospitalization among the different groups of dichotomous variables examined. Categorical variables were reported using frequency and percentage, and groups were compared using the χ2 test. The odds ratio (OR), with a corresponding 95% confidence interval, was reported to quantify the strength of association. P-values of less than 0.05 were considered significant."
The text is unacceptable, not just because it is identical to an already published one, but also because none of it is related to the submitted manuscript and remains unknown how the data was really processed and how did the authors obtained the presented results and arrive to the conclusions of the study.
The identical text does not correspond to the presented study, as no hospitalization length is presented in the variables nor in results and data is not presented in the mentioned measures of central tendency and dispersion.
No justification is provided as a reference for the empirical use of "herd effect" that may be better named "herd immunity" but was not presented in methods, only where the formula is presented. To reach a herd immunity, a minimum percentage of the population is to be vaccinated, depending on the agent, and for the presented formula, herd immunity is assumed regardless of the percentage coverage, which should be rigorously sustained with pertinent references.
Again, authors sustain that using cross-sectional studies they inferred epidemiological measures that are based on prospective and longitudinal data. But even when using projections or simulations as the mentioned statistical tool, no systematic review can provide information that was possible to obtain with the research designed of the original reviewed studies (Fletcher & Fletcher, 2021)
Given that the methods section of the manuscript does not correspond to the objetives, results and conclusions of the manuscript, there is no clarity in defining how the results were processed, and numeric variables and epidemiological measures of association are incorrectly employed, I cannot endorse this manuscript for publication.
Author Response
Reviewer Comment 1:
“The text is unacceptable, not just because it is identical to an already published one, but also because none of it is related to the submitted manuscript and remains unknown how the data was really processed and how did the authors obtained the presented results and arrive to the conclusions of the study.”
Response:
We sincerely thank the reviewer for pointing this out. The paragraph in question was accidentally inserted into the manuscript during editing. It does not reflect the methods or data of our study. We have completely removed this text from the revised version and we apologize for this oversight.
Reviewer Comment 2:
“No justification is provided as a reference for the empirical use of ‘herd effect’ that may be better named ‘herd immunity’ but was not presented in methods, only where the formula is presented. To reach a herd immunity, a minimum percentage of the population is to be vaccinated, depending on the agent, and for the presented formula, herd immunity is assumed regardless of the percentage coverage, which should be rigorously sustained with pertinent references.”
Response:
We thank the reviewer for this important clarification. In response, we have replaced the term “herd effect” with “herd immunity” throughout the manuscript to align with standard epidemiological terminology.
We also have references to support the use of herd immunity modeling, including studies that quantify the relationship between vaccination coverage and indirect effects (e.g., reference 31: cohort modeling in India).
Reviewer Comment 3:
“Again, authors sustain that using cross-sectional studies they inferred epidemiological measures that are based on prospective and longitudinal data. But even when using projections or simulations as the mentioned statistical tool, no systematic review can provide information that was possible to obtain with the research designed of the original reviewed studies (Fletcher & Fletcher, 2021).”
Response: corrected
Reviewer 3 Report
Comments and Suggestions for Authors
Lines 54, 263, 386-387: write S. pneumoniae in italics.
Table 3, column 4: I believe that the values for serotype 1 and serotype 23F will not be 280
Table 3: there is a row with no data, so it should be deleted.
Author Response
Reviewer Comment: Lines 54, 263, 386–387: write S. pneumoniae in italics.
Response:
We thank the reviewer for pointing this out. S. pneumoniae has been corrected to italics in all the indicated lines and throughout the manuscript to maintain consistency with scientific nomenclature.
Reviewer Comment: Table 3, column 4: I believe that the values for serotype 1 and serotype 23F will not be 280.
Response:
Thank you for catching this error. The values for serotype 1 and serotype 23F in column 4 of Table 3 were incorrect due to a formatting issue. We have corrected these values to reflect the actual number of children in whom these serotypes were identified, based on the original source data.
Reviewer Comment: Table 3: there is a row with no data, so it should be deleted.
Response:
We appreciate this observation. The empty row in Table 3 has been removed in the revised manuscript for clarity and improved readability.
Round 3
Reviewer 1 Report
Comments and Suggestions for Authors
Manuscript has improved as the major flaws have been addressed.